# Bridging the Space Gap: Unifying Geometry Knowledge Graph Embedding with Optimal Transport

## ABSTRACT

Knowledge Graph Embedding (KGE) is a critical field aiming to transform the elements of knowledge graphs into continuous spaces, offering great potential for structured data representation. In contemporary KGE research, the utilization of either hyperbolic or Euclidean space for knowledge graph modeling is a common practice. However, Knowledge graphs encompass diverse geometric data structures, including chains and hierarchies, whose hybrid nature exceeds the capacity of a single embedding space to capture effectively. This paper introduces a groundbreaking and highly effective approach called *Unified Geometry Knowledge Graph Embedding* (UniGE) to address the challenge of representing diverse geometry data in KGs. UniGE stands out as the pioneering KGE methodology that seamlessly integrates knowledge graph embeddings in both Euclidean and hyperbolic geometric spaces. We introduce an embedding alignment method and fusion strategy, which harnesses optimal transport techniques and Wasserstein barycenter method. Furthermore, we offer a comprehensive theoretical analysis to substantiate the superiority of our approach, as evident from a more robust error bound. To substantiate the strength of UniGE, we conducted comprehensive experiments on three benchmark datasets. The results consistently demonstrate that UniGE outperforms state-of-the-art methods, aligning with the conclusions drawn from our theoretical evaluation.

## KEYWORDS

Knowledge Graph Embedding, Optimal Transport, Euclidean space, Hyperbolic Space

**ACM Reference Format:**
Anonymous Author(s). 2018. Bridging the Space Gap: Unifying Geometry Knowledge Graph Embedding with Optimal Transport. In *Proceedings of Make sure to enter the correct conference title from your rights confirmation emai (ACM'The Web Conference)*. ACM, New York, NY, USA, 9 pages. https://doi.org/XXXXXXX.XXXXXXX

## 1 INTRODUCTION

The creation and utilization of Knowledge Graphs (KGs) have garnered significant attention from academia and industry alike. Its applications have rapidly evolved, spanning from recommender systems [7], dialogue generation [10], to question-answering systems [6]. Despite its efficacy in structured data representation, the

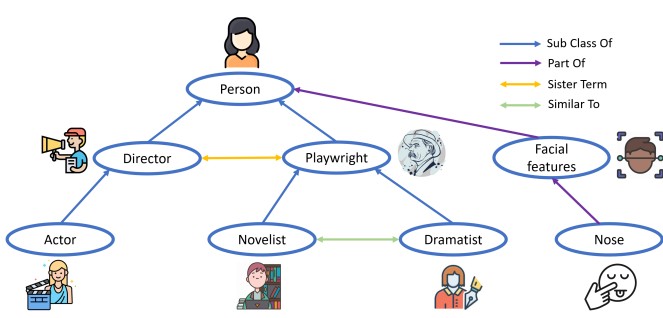

**Figure 1: A knowledge graph contains multiple distinct hierarchies (e.g.,subClassOf and partOf ) and non-hierarchical relations (e.g.,similarTo and sisterTerm)**

fundamental symbolic nature of KG triples poses a challenge for manipulation. To address this issue, the KGE method has emerged as a practical solution in recent years. The primary aim of KGE is to embed KG components (such as entities and relations) into a continuous vector space, which preserves the intrinsic KG structure while streamlining operations. Furthermore, KGEs can explicitly capture the similarity between entities and relations by measuring the similarity of their low-dimensional embeddings.

Recent significant KGE works have predominantly viewed knowledge graph modeling through the lenses of Euclidean space $\mathbb{E}$ and ComplEx space $\mathbb{C}$, introducing models like RotatE [23] and ComplEx [25]. These studies have shown that modeling in Euclidean space can provide a suitable representation of entities and relationships, particularly for the chain structure in knowledge graphs. However, it has become evident that training KGE in Euclidean space falls short of adequately capturing the hierarchical structural information present in knowledge graphs [1]. In response to this limitation, recent KGE works have ventured into the use of hyperbolic embedding space. For instance, Kolyvakis et al. [11] extend translational models to the hyperbolic space, enabling a more accurate reflection of the topological properties of knowledge bases. Additionally, Sun et al. [22] propose a hyperbolic relational graph neural network for KGE, which captures knowledge associations with a hyperbolic transformation.

Figure 1 illustrates a typical example of a KG, showcasing a heterogeneous graph structure with both hierarchy and chain components, along with interactive relationships between them. Emerging research [1, 23] has highlighted that a single geometric space can effectively represent a specific type of KG. For example, Euclidean space is well-suited for accurately representing chain-structured KGs. Conversely, hyperbolic space is better equipped to handle the diverse topological structures often associated with hierarchical real-world KGs. These relations encompass interactions between entities at the same level of the hierarchy, as well as non-hierarchical relationships such as "similar to" and "sister team." Consequently, to

model the knowledge graph more effectively, it becomes imperative to utilize a combination of Euclidean and hierarchical structures for modeling. Motivated by these findings, we develop a unified geometric space representation for KG modeling, capable of better representing the data's heterogeneous structural aspects in the KG. Moreover, this unified representation leverages the strengths of both Euclidean and hyperbolic spaces.

In this paper, we introduce a novel approach, *Unified Geometry Knowledge Graph Embedding* (UniGE), which unify the embedding of KG components in both Euclidean and hyperbolic geometric spaces. Specifically, we formulate the fusion of KGE in Euclidean and hyperbolic spaces as an optimal transport problem. Optimal transport facilitates the movement of heterogeneous geometric space embeddings to a unified, aligned space by minimizing the Wasserstein distance between different distributions. Leveraging the Wasserstein barycenter, we enable seamless information communication between Euclidean and hyperbolic geometric spaces. Additionally, we provide theoretical evidence that the Wasserstein distance can be used to limit the divergence between the distributions of the respective source geometric spaces and the unified space, ultimately leading to a lower error bound than that observed in Euclidean and hyperbolic geometric spaces. Furthermore, empirical experiments substantiate that UniGE significantly outperforms state-of-the-art methods.

Our contributions are summarized as follows:

- We are the pioneers in exploiting KGE with a unified embedding space, enabling more comprehensive modeling of hierarchical and non-hierarchical structures in KGs.
- We present a novel method, UniGE, based on optimal transport, which effectively addresses geometry space heterogeneity by reducing the Wasserstein distance between various embedding space distributions.
- Theoretically, we demonstrate that our proposed method can achieve a lower error bound than that observed in Euclidean and hyperbolic geometric spaces.
- Extensive experiments show that UniGE achieves state-of-the-art performance on three well-established knowledge graph completion benchmarks (WN18RR, FB15K237, and YAGO3-10).

## 2 RELATED WORK

Recent advancements in KGE have greatly benefited from the exploration of geometric features and more complex spaces for precise modeling. In this section, we provide an overview of KGE models that leverage geometric characteristics to preserve the structural integrity of KGs. Additionally, we delve into the general concept of the optimal transport problem. To the best of our knowledge, our approach marks the first attempt to unify geometry within the domain of KGE using optimal transport.

### 2.1 KGE in Euclidean Space

The domain of Euclidean space has been a focal point of research in knowledge graph embedding. Notable models such as TransE [2], which interprets relationships as translations on low-dimensional entity embeddings, and its variants, including those by [9, 15, 26], have explored various forms of relational patterns. RotatE [23],

a model developed by Sun et al. [23], excels in modeling a wide range of relation patterns, including symmetry, antisymmetry, inversion, and composition. Other KGE models, such as ComplEx [25] and DistMult [28], employ element-wise multiplication to handle a broad spectrum of binary relations. Models like RESCAL [19] and QuatE [29] utilize angle transformation to model relational patterns, while MuRE [1] adopts a diagonal relational matrix. While these methods have demonstrated remarkable performance, they have limitations when it comes to encoding complex hierarchical and chaining components due to their reliance on Euclidean space representations. Our approach leverages optimal transport to incorporate information from hyperbolic space to address this challenge.

### 2.2 KGE in Hyperbolic Space

Hyperbolic space has recently garnered considerable attention due to its potential to represent symbolic data more effectively by capturing hierarchical structure [13, 21]. Several pioneering efforts in hyperbolic KGE have emerged in contrast to the Euclidean space. Balazevic et al. [1] introduces the MuRP model, which utilizes Möbius matrix-vector multiplication and Möbius addition operations for embedding multi-relational KG data in the Poincaré ball model of hyperbolic space. Kolyvakis et al. [11] extends translational models into hyperbolic space to better capture the topological characteristics of knowledge bases. Sun et al. [22] introduces a hyperbolic relational graph neural network for KGE, incorporating a hyperbolic transformation to capture knowledge relationships. Chami et al. [4] introduces a class of hyperbolic KGE models that simultaneously capture hierarchical and logical patterns, including RotH, RefH, and AttH. Our model is distinctive in its ability to unify geometry in both Euclidean and Hyperbolic KGE models, delivering competitive performance compared to existing methods.

### 2.3 Optimal Transport

Optimal transport, as discussed by Monge [17], presents a versatile tool with potential applications in machine learning, notably for finding the optimal path between two targets. Recent studies have harnessed the power of optimal transport in various ways. For example, Nouri [20] incorporats syntactic and semantic information into similarity computations between source and converted text using optimal transport. Tang et al. [24] assesss the semantic coverage of summaries in relation to the original document through optimal transport. Additionally, Li et al. [14] employs optimal transport to reduce exposure bias by matching sequences generated during training and testing. Our approach leverages optimal transport to incorporate information from both Euclidean and hyperbolic space for KGE.

## 3 PRELIMINARIES AND BACKGROUND

This section lays the foundation by introducing the prerequisites of KGE, as well as the fundamentals of Euclidean and Hyperbolic geometry. It elucidates the essential components essential for a comprehensive understanding of our model. In particular, our model seeks to establish a unified framework for geometry within the realm of KGE, and we provide the requisite contextual background.

**Table 1: Summary of characteristic properties and operations in Euclidean and Hyperbolic space**

|  | Euclidean Space | Hyperbolic Space |
|---|---|---|
| **Curvature** | $= 0$ | $< 0$ |
| **Manifold $\mathbb{M}^d$** | $\mathbb{R}^d$ | $\{x \in \mathbb{R}^d : \|x\| < -\frac{1}{K}\}$ |
| **Sum of triangle angles** | $\pi$ | $< \pi$ |
| **Exponential map $exp_x(v)$** | $x + v$ | $x \oplus_k (tanh(\sqrt{\|K\|}\frac{\lambda_x^k \|v\|}{2})\frac{v}{\sqrt{\|K\|}\|v\|})$ |

## 3.1 Knowledge Graph Embedding

In the context of KGE, a set of triples $(h, r, t) \in \mathcal{E} \subseteq \mathcal{V} \times \mathcal{R} \times \mathcal{V}$ is defined, where $\mathcal{V}$ and $\mathcal{R}$ represent the sets of entities and relationships within the KGs, respectively. To model the KG structure, we employ an embedding space denoted as $\mathcal{U}$, which encompasses both Euclidean and hyperbolic geometry embeddings. Entities $v \in \mathcal{V}$ are embedded into representations $e_v \in \mathcal{U}^{d_{\mathcal{V}}}$, and relationships $r \in \mathcal{R}$ are mapped to embeddings $r_r \in \mathcal{U}^{d_{\mathbb{R}}}$. Furthermore, our dataset is segregated into two subsets: $\mathcal{E}_{Train}$ and $\mathcal{E}_{Test}$ triples. The training for KGE involves optimizing a scoring function $S : \mathcal{V} \times \mathcal{R} \times \mathcal{V} \longrightarrow \mathbb{R}$, which quantifies the likelihood of tuples. The scoring function, $S(\cdot, \cdot, \cdot)$, is learned by utilizing triples from $\mathcal{E}_{Train}$, and subsequently, the trained embeddings are employed to assess the performance of triples in $\mathcal{E}_{Test}$.

## 3.2 Euclidean and Hyperbolic Embedding Space

Non-Euclidean embedding spaces, characterized by their curvature in contrast to the zero-curvature Euclidean embedding space, exhibit a range of distinct features. The Riemannian manifolds $\mathbf{M}^d$ encompass the Euclidean embedding space $\mathbb{E}^d$ and the hyperbolic spaces $\mathbb{H}^d$, where each point $x \in \mathbf{M}^d$ possesses a tangent space denoted as $(T_x \mathbf{M}^d)^d$, offering a local approximation of $\mathbf{M}^d$ in the vicinity of $x$. Notably, every Riemannian manifold is equipped with a Riemannian metric distance that quantifies the geodesic separation between any two points within the manifold. Additionally, the curvature $K$ of the space varies: Euclidean spaces maintain $K_E = 0$ and are frequently utilized in numerous publications, while hyperbolic spaces exhibit negative curvature ($K_H < 0$) and are more adept at capturing power-law patterns, as discussed by Nickel and Kiela [18]. Table 1 provides a concise summary of the key attributes distinguishing Euclidean and hyperbolic spaces.

**Hyperbolic distance.** In hyperbolic space, the distance between embedding $\mathbf{u}$ and $\mathbf{v}$ is expressed as follows:

$$D_H(\mathbf{u}, \mathbf{v}) = arccosh(1 + 2\frac{\|\mathbf{u} - \mathbf{v}\|^2}{(1 - \|\mathbf{u}\|^2)(1 - \|\mathbf{v}\|^2)}),$$

from the distance formula, we can deduce that as points move away from the origin towards the boundary of the ball, the distance grows exponentially. This phenomenon results in a significantly larger volume of space available for KGE.

**Hyperbolic addition.** The vector addition in hyperbolic space is defined by the Möbius addition :

$$\mathbf{u} \oplus \mathbf{v} = \frac{(1 + 2\langle\mathbf{u}, \mathbf{v}\rangle + \|\mathbf{v}\|^2)\mathbf{u} + (1 - \|\mathbf{u}\|^2)\mathbf{v}}{1 + 2\langle\mathbf{u}, \mathbf{v}\rangle + \|\mathbf{u}\|^2\|\mathbf{v}\|^2}.$$

**Exponential and Logarithmic maps.** Hyperbolic embeddings are initially projected into the tangent space at $\mathbf{0}$ using the logarithmic map. Subsequently, operations similar to those in Euclidean space are performed, and the results are finally projected back onto the manifold using the exponential map. The mathematical definitions of these two mappings are as follows:

$$exp_{\mathbf{0}}(\mathbf{u}) = tanh(\|\mathbf{u}\|)\frac{\mathbf{u}}{\|\mathbf{u}\|},$$

$$log_{\mathbf{0}}(\mathbf{u}) = tanh^{-1}(\|\mathbf{u}\|)\frac{\mathbf{u}}{\|\mathbf{u}\|}.$$

Matrix-vector multiplication in the hyperbolic embedding space can be performed using the Möbius map:

$$\mathbf{M} \odot \mathbf{u} = exp_{\mathbf{0}}(\mathbf{M}log_{\mathbf{0}}(\mathbf{u})).$$

## 4 METHODOLOGY

Our model proposes the development of a unified KGE in geometry, seamlessly integrating the advantages offered by the Euclidean space with the structural benefits of the hyperbolic space. In contrast to existing KGE models, which are limited to representing data in a specific geometric context [2, 4, 23], we introduce a comprehensive model termed UniGE. Utilizing optimal transport techniques, our model fuses the representations of both Euclidean and hyperbolic spaces, harnessing the strengths of both paradigms. In Section 5, we provide an in-depth theoretical analysis.

### 4.1 Overview

The schematic representation of our proposed model is illustrated in Figure 2. Our model comprises a foundational structure that integrates models based on both Euclidean and hyperbolic spaces, a KGE alignment dynamics learner employing optimal transport principles, and a KGE fusion module designed to harmonize embeddings from both spatial contexts. Notably, the embedding alignment dynamics learner and the fusion module constitute the central functional components of our model.

### 4.2 Representations for Entities and Relations

For KGE, it is imperative to represent entities and relationships as vectors that encapsulate meaningful semantic information. Prior research has primarily concentrated on modeling relationships and entities within a single embedding space, which typically encompasses both Euclidean and hyperbolic geometries. However, KGs often contain data with diverse structural attributes, necessitating the amalgamation of multiple embedding spaces to effectively model them.

 

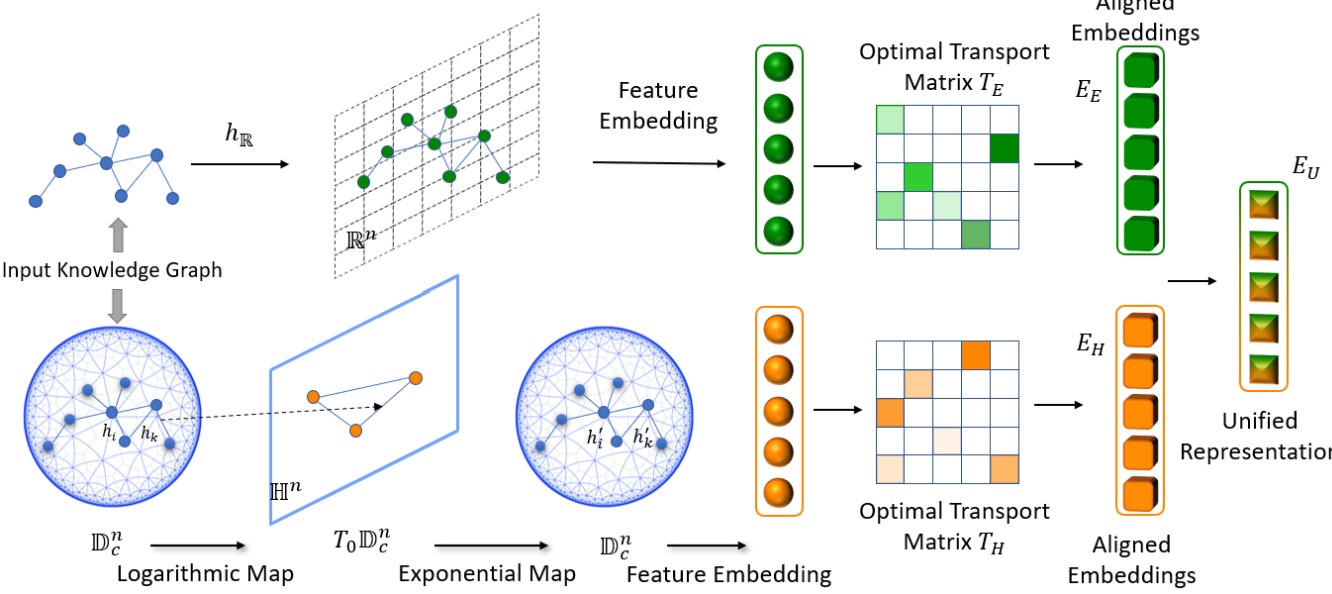

**Figure 2: Our model encompasses a backbone consisting of both euclidean (top) and hyperbolic (bottom) space models, a KGE alignment dynamics learner based on optimal transport, and a KGE fusion module to unify both embedding spaces.**

The embedding methods employed within the Euclidean and hyperbolic geometry spaces differ. Specifically, for an entity denoted as $e^H \in \mathbf{B}_c^d$, it corresponds to the hyperbolic embeddings of entity $e$. For a given relation $r$, we define two hyperbolic relation vectors, $r^H$ and $r'^H \in \mathbf{B}_c^d$, representing two translation operations. These hyperbolic embeddings are parameterized using a dimensional vector $\hat{r}$, which subsequently defines a givens Rotation operation with a block-diagonal matrix structure:

$$Rot(\hat{r}) = diag(G(\hat{r}_1, \hat{r}_2), \ldots, G(\hat{r}_{d-1}, \hat{r}_d)). \qquad (1)$$

Then, for a triple $(h, r, t)$, the scoring function $F_H$ is defined as:

$$
\begin{aligned}
Q_H^c(h, r) &= Rot(\hat{r}) \otimes_c (h^H \oplus_c r^H) \oplus_c r'^H, \\
D_H^c(q, t) &= -\frac{2}{\sqrt{c}} arctanh(\sqrt{c}\| - q \oplus_c t^H\|)^2, \qquad (2) \\
F_H(h, r, t) &= D_H^{c_r}(Q_H^c(h, r), t) + b_h + b_t,
\end{aligned}
$$

where $c_r > 0$ is the relation-specific curvature parameter and $b_e$ are entity biases that act as margins in the scoring function [1, 4].

In addition, the embedding methods we used in Euclidean Space are similar to the one described above, but it is based on Euclidean space, and their scoring function is defined as:

$$F_E(h, r, t) = -\|(Rot(\hat{r})h + r) - t\|^2 + b_h + b_t, \qquad (3)$$

where $h, r, t \in \mathbb{R}^d$. Without complex hyperbolic calculations, $F_E$ can be computed in linear time of the embedding dimensions.

### 4.3 Knowledge Graph Embedding Alignment

Our model integrates the representation of Euclidean and hyperbolic spaces through optimal transport, leveraging the advantages of both. Since the embedding distributions in the two spaces do not

align directly, combining them directly would disrupt the original pattern relationships. Therefore, we employ optimal transport to achieve embedding alignment.

We provide a brief explanation of the transportation problem and how to interpret the total transportation cost as an embedding distance measure. A transportation problem consists of three key components: the initial and final states of transportation and a cost matrix. Typically, the two states are represented in Euclidean and hyperbolic spaces, denoted as $e^e \in \mathbf{R}^n$ and $e^h \in \mathbf{H}^m$, where each dimension corresponds to a specific location with a non-negative quantity. Following the aforementioned steps, we use $\mathbf{u} = \{u_i\}_{i=1}^n$ and $\mathbf{v} = \{v_j\}_{j=1}^m$ to represent the probabilistic simplexes of Euclidean and hyperbolic knowledge graph embeddings, respectively. We denote $\Omega_e = \sum_{i=1}^n u_i \delta_{e^e}$ and $\Omega_h = \sum_{j=1}^m v_j \delta_{e^h}$ as discrete distributions of Euclidean and hyperbolic knowledge embeddings, where $\delta$ represents the Dirac function. The cost matrix $\mathbf{M} \in \mathbb{R}^{n \times m}$ encodes the unit transportation cost from location $i$ to $j$ in $\mathbf{M}_{i,j}$.

In this context, we seek a transportation plan to move from $e^e$ to $e^h$ while minimizing the overall cost. Using the above notations, the optimization problem can be formulated as follows:

$$
\begin{aligned}
&\underset{T \in \mathbb{R}^{n \times m}}{minimize} \sum_{i,j} \mathbf{T}_{i,j} \mathbf{M}_{i,j} \\
&s.t. \sum_{j=1}^n t_{ij} = \mathbf{u}, \sum_{i=1}^n t_{ij} = \mathbf{v}, \qquad (4) \\
&t_{ij} \ge 0, \forall i, j \in 1, \ldots, N,
\end{aligned}
$$

where the first two constraints indicate the quantity requirements for both suppliers and customers, and the last constraint proves a non-negative order quantity. Mathematically, this OT problem

is to find a joint distribution $\mathbf{T}$ concerning a cost $\mathbf{M}$ of which the marginal distribution is in the Euclidean and hyperbolic spaces. In particular, Wasserstein distance can be defined as:

$$\mathcal{W}_p^p(\Omega_e, \Omega_h) = \min_{T \in \prod(\Omega_e, \Omega_h)} \sum_{i=1}^{n} \sum_{j=1}^{m} \mathbf{T}_{i,j} \mathbf{M}_{i,j}. \tag{5}$$

It can be viewed as the distance between the two probability distributions $\Omega_e$ and $\Omega_h$, if they are normalized, in line with the cost $\mathbf{M}$.

In our model, an important step is introduced to approximate the entities' semantic distributions in the Euclidean and hyperbolic spaces. We define the unit transportation cost between two KGE in the two different spaces by measuring their semantic similarity. Intuitively, the more semantically dissimilar a pair of knowledge graph embeddings are, the higher the "transport cost" of transporting one knowledge graph embedding to another. Given two knowledge graph embedding models in different spaces, define $e_i^e$ and $e_j^h$ to represent the feature embedding in the two different spaces. The transport cost from Euclidean space to hyperbolic spaces can be written as follows:

$$M_{ij} = \|e_i^e - e_j^h\|_2. \tag{6}$$

## 4.4 Knowledge Graph Embedding Fusion

After obtaining the transport matrix $T$, the hyperbolic knowledge graph embedding $e^h$ can be transformed into target-aligned embedding $\hat{e}^h$ with the barycenter-based strategies:

$$\begin{aligned} \hat{e}^e &= e^e \times diag(\frac{1}{u}) \times (T^T + \Delta_T), \\ \hat{e}^h &= e^h \times diag(\frac{1}{v}) \times (T^T + \Delta_T). \end{aligned} \tag{7}$$

After obtaining the target-aligned knowledge graph embedding, the transported Euclidean and hyperbolic knowledge embeddings are in the same space, where $\Delta_T$ is an adjustable transport parameter. We obtain the aligned knowledge graph embedding in the different spaces from the above equation. Then the next step is to fuse the aligned different knowledge graph embedding with Barycenters strategy:

$$e^u = \frac{1}{2} \min_{e^e} \sum_{i \in \{e,h\}} \lambda_i \mathcal{W}, (\hat{e}^i, e^e), \tag{8}$$

where $e^u$ is the unified representation; $\lambda_i$ represents the weight.

## 4.5 Loss Function and Training

After performing the above knowledge graph embedding alignment and fusion, we can obtain the loss function from the score function. To optimize the parameters, we train the model by minimizing the following loss:

$$L = \sum log(1 + exp(-Y_{label}F(h, r, t))), \tag{9}$$

where $Y_{label} \in \{-1, 1\}$ denotes the label of the triple $(h, r, t)$. In the training procedure, we adopt the negative sampling strategies (e.g., uniform sampling or Bernoulli sampling [26]).

## 5 THEORETICAL ANALYSIS

This section shows a theoretical analysis showing that unified KGE preserves comprehensiveness.

The target error [5] is an indicator to measure the distribution difference between the final unified KGE and the respective KGE. Supposing $\mathcal{X}, \mathcal{Y}, \mathcal{Z}$ are represented for entities, KGE, and labels of triples, respectively. In addition, the euclidean and hyperbolic embedding of the entity is always assumed to satisfy the proper function $f^* : \mathcal{Z} \longrightarrow \mathcal{Y}$ in KGE problems. The proper scoring function $f^*$ can not be viewed, so we usually choose a prediction function $f$ from a hypothesis class $\mathcal{F}$ for substitution.

Afterward, there is an approximation error between the hypothesis $f$ and the true score function $f^*$ under the distribution $\Omega$, we measure it here with the taget error :

$$err_\Omega(f, f^*) \stackrel{def}{=} \mathbb{E}_{x \in \Omega}[|f(x) - f^*(x)|]. \tag{10}$$

To simplify the proof, we denote $err_\Omega(f, f^*)$ as $err_\Omega(f)$. In addition, we utilize the Wasserstein distance $\mathcal{W}_1(\cdot, \cdot)$ to relate the source distribution and unified distribution.

We can make inferences about the unified embedding space based on the above definitions and assumptions.

THEOREM 1 (THE MINIMIZED BOUND ON THE ERROR). *Supposing that $\Omega_E, \Omega_H$ and $\Omega_U$ represent the distribution of the euclidean, hyperbolic and unified embedding, and the hypotheses $f, f^* \in \mathcal{F}$ are all L-Lipschitz continuous for some constant L. Then the following inferences hold for every hypothesis $f, f^* \in \mathcal{F}$:*

$$err_U(f) \leq \min_{i \in \{E,H\}} \{err_i(f) + \mathcal{W}_1(\Omega_i, \Omega_U)\}, \tag{11}$$

*where $\mathcal{W}_1(\cdot, \cdot)$ is the 1-Wasserstein distance, $err_*(f)$ is the target error at the respective knowledge graph embedding space $*$.*

PROOF. Due to $f \in \mathcal{F}$ is L-Lipschitz continuous , $\forall x, y \in \mathcal{D}$, there is $|f(x) - f(y)| \leq Ld(x, y)$. Given the L-Lipschitz continuous hypotheses $f, f' \in \mathcal{F}$, we can obtain the following formula with the triangle inequality:

$$\begin{aligned} |f(x) - f'(x)| &\leq |f(x) - f(y)| + |f(y) - f'(x)| \\ &\leq |f(x) - f(y)| + |f(y) - f'(y)| \\ &\quad + |f'(y) - f'(x)|. \end{aligned} \tag{12}$$

Assuming that $d(x, y)$ represents a measure for the distance between $x$ and $y$, $\forall x, y \in \mathcal{X}$, we can obtain:

$$\begin{aligned} |f(x) - f'(x)| &- |f(y) - f'(y)| \\ &\leq |f(x) - f(y)| + |f'(y) - f'(x)| \\ &\leq 2Ld(x, y). \end{aligned} \tag{13}$$

We can infer that given the euclidean and hyperbolic embedding space distribution,

$$\begin{aligned} err_U(f, &f^*) - err_H(f, f^*) \\ &= \mathbb{E}_{x \sim \Omega_U}[|f(x) - f'(x)|] - \mathbb{E}_{x \sim \Omega_H}[|f(x) - f'(x)|] \\ &\leq \sup_{\|f\|_2 \leq 2L} \mathbb{E}_{\Omega_U}[f(x)] - \mathbb{E}_{\Omega_H}[f(x)] \\ &\leq 2L\mathcal{W}_1(\Omega_H, \Omega_U). \end{aligned} \tag{14}$$

**Table 2: Link prediction results (%) on WN18RR, FB15k-237 and YAGO3-10 for low-dimensional embeddings ($d$ = 32) in the filtered setting. The first group of models are Euclidean models, and the second groups are non-Euclidean models baseline. All results are taken from [27]. The best score and best baseline are in bold and underlined, respectively.**

|  | WN18RR | | | | FB15k-237 | | | | YAGO3-10 | | | |
|---|---|---|---|---|---|---|---|---|---|---|---|---|
| Model | MRR | H@1 | H@3 | H@10 | MRR | H@1 | H@3 | H@10 | MRR | H@1 | H@3 | H@10 |
| TransE | 36.6 | 27.4 | 43.3 | 51.5 | 29.5 | 21.0 | 32.2 | 46.6 | - | - | - | - |
| RotatE | 38.7 | 33.0 | 41.7 | 49.1 | 29.0 | 20.8 | 31.6 | 45.8 | - | - | - | - |
| ComplEx | 42.1 | 39.1 | 43.4 | 47.6 | 28.7 | 20.3 | 31.6 | 45.6 | 33.6 | 25.9 | 36.7 | 48.4 |
| MuRE | 45.8 | 42.1 | 47.1 | 52.5 | 31.3 | 22.6 | 34.0 | 48.9 | 28.3 | 18.7 | 31.7 | 47.8 |
| MuRP | 46.5 | 42.0 | 48.4 | 54.4 | 32.3 | 23.5 | 35.3 | 50.1 | 23.0 | 15.0 | 24.7 | 39.2 |
| RotH | 47.2 | 42.8 | 49.0 | 55.3 | 31.4 | 22.3 | 34.6 | 49.7 | 39.3 | 30.7 | 43.5 | 55.9 |
| RefH | 44.7 | 40.8 | 46.4 | 51.8 | 31.2 | 22.4 | 34.2 | 48.9 | 38.1 | 30.2 | 41.5 | 53.0 |
| AttH | 46.6 | 41.9 | 48.4 | 55.1 | 32.4 | 23.6 | 35.4 | 50.1 | 39.7 | 31.0 | 43.7 | 56.6 |
| UltraE | 48.8 | 44.0 | 50.3 | 55.8 | 33.8 | 24.7 | 36.3 | 51.4 | 40.5 | 31.8 | 44.7 | 57.2 |
| UniGE(Ours) | **49.1** | **44.7** | **51.2** | **56.3** | **34.3** | **25.7** | **37.5** | **52.3** | **41.2** | **32.5** | **45.1** | **57.9** |

Then,we can derive the following statement:

$$err_U(f) \le err_H(f) + 2L\mathcal{W}_1(\Omega_H, \Omega_U). \tag{15}$$

Similarly,we can derive the following formula:

$$err_U(f) \le err_E(f) + 2L\mathcal{W}_1(\Omega_E, \Omega_U). \tag{16}$$

Obviously,

$$err_U(f) \le \min_{i \in \{E,H\}} \{err_i(f) + \mathcal{W}_1(\Omega_i, \Omega_U)\}. \tag{17}$$

□

From the above proof, it is possible to obtain a reasonably unified KGE, which the target error can be smaller than any involved KGE error.

## 6 EXPERIMENTS

### 6.1 Experiment Setup

*6.1.1 Datasets.* Our approach is rigorously evaluated on the task of link prediction, utilizing three well-established competitive benchmarks: WN18RR [2], FB15K237 [2], and YAGO3-10 [16]. Each of these datasets offers distinct challenges and characteristics for assessing the efficacy of our method.

- **WN18RR**, a subset of WN18, is known for its primary relation patterns of symmetry/antisymmetry and composition.
- **FB15K237**, derived from FB15K, retains its distinctiveness by excluding inverse relations.
- **YAGO3-10**, a subset of YAGO3, predominantly comprises symmetry/antisymmetry and composition patterns.

To ensure a consistent evaluation framework, we adopt the standard dataset split configuration of training, validation, and testing sets, as outlined in the work by Sun et al. [23]. Furthermore, we apply a standard data augmentation technique by incorporating inverse relations into the baseline datasets, following the protocol established by Lacroix et al. [12] ,and strictly adhere to the established evaluation protocol in the filtered setting [2]. Notably, for evaluation, we exclude all genuine triplets already present in the

KG. This practice is vital because penalizing low ranks for triplets that are already part of the KG would be inconsistent with the task's objectives.

For a more comprehensive description of the datasets, we direct readers to the Appendix A. We also commit to releasing the source code upon acceptance for the benefit of the research community.

*6.1.2 Evaluation Metrics.* In each experimental scenario, we evaluate our approach by ranking the test triplets among all possible triplets, where entities that were initially masked are substituted with entities from the KG. To evaluate our method's performance, we report key evaluation metrics, including Hit@n (with n values of 1, 3, and 10) and the Mean Reciprocal Rank (MRR).

*6.1.3 Hyperparameters.* In this study, we meticulously explore a range of hyperparameters tailored to each KG to fine-tune our model's performance. Specifically, we experiment with two distinct batch sizes, 500 and 1000, and three learning rates, 3e-3, 5e-3, and 7e-3, all of which are evaluated on the validation sets. We maintain a fixed negative sampling size of 50 and establish a maximum number of training epochs set at 1000. During training, we employ an early stopping mechanism, intervening if the validation MRR ceases to exhibit improvement after 100 epochs. Our choice of optimizers is tailored to each dataset: for WN18RR and YAGO3-10 datasets, we opt for the Adam optimizer, whereas for the FB15k-237 dataset, we employ Adagrad. Notably, for certain baseline methods, we report results as outlined in their respective original papers for reference.

### 6.2 Baseline

- **Euclidean models.** 1) TransE [2], the first translational model; 2) RotatE [23], a rotation model in a complex space; 3) DistMult [28], a multiplicative model with a diagonal relational matrix; 4) ComplEx [25], an extension of DisMult in a complex space; 5) MuRE [1], a Euclidean model with a diagonal relational matrix.

**Table 3: Link prediction results (%) on WN18RR, FB15k-237 and YAGO3-10 for high-dimensional embeddings (best for $d \in \{200, 400, 500\}$ ) in the filtered setting. All results are taken from [27]. The best score and best baseline are in bold and underlined, respectively.**

| Model | WN18RR | | | | FB15k-237 | | | | YAGO3-10 | | | |
|---|---|---|---|---|---|---|---|---|---|---|---|---|
| | MRR | H@1 | H@3 | H@10 | MRR | H@1 | H@3 | H@10 | MRR | H@1 | H@3 | H@10 |
| TransE | 48.1 | 43.3 | 48.9 | 57.0 | 34.2 | 24.0 | 37.8 | 52.7 | - | - | - | - |
| RotatE | 47.6 | 42.8 | 49.2 | 57.1 | 33.8 | 24.1 | 37.5 | 53.3 | 49.5 | 40.2 | 55.0 | 67.0 |
| ComplEx | 48.0 | 43.5 | 49.5 | 57.2 | 35.7 | 26.4 | 39.2 | 54.7 | 56.9 | 49.8 | 60.9 | 70.1 |
| MuRE | 47.5 | 43.6 | 48.7 | 55.4 | 33.6 | 24.5 | 37.0 | 52.1 | 53.2 | 44.4 | 58.4 | 69.4 |
| MuRP | 48.1 | 44.0 | 49.5 | 56.6 | 33.5 | 24.3 | 36.7 | 51.8 | 35.4 | 24.9 | 40.0 | 56.7 |
| RotH | 49.6 | 44.9 | 51.4 | 58.6 | 34.4 | 24.6 | 38.0 | 53.5 | 57.0 | 49.5 | 61.2 | 70.6 |
| RefH | 46.1 | 40.4 | 48.5 | 56.8 | 34.6 | 25.2 | 38.3 | 53.6 | 57.6 | 50.2 | 61.9 | 71.1 |
| AttH | 48.6 | 44.3 | 49.9 | 57.3 | 34.8 | 25.2 | 38.4 | 54.0 | 56.8 | 49.3 | 61.2 | 70.2 |
| UltraE | 50.1 | 45.0 | 51.5 | **59.2** | **36.8** | **27.6** | **40.0** | **56.3** | 58.0 | 50.6 | 62.3 | 71.1 |
| UniGE(Ours) | **50.2** | **45.5** | **52.0** | **59.2** | 35.7 | 26.4 | 39.1 | 55.9 | **58.3** | **51.2** | **62.7** | **71.5** |

- **Non-Euclidean models.** 1) MuRP [1], a hyperbolic model with a diagonal relational matrix; 2) MuRS, a spherical analogy of MuRP; 3) RotH/RefH [4], a hyperbolic embedding with rotation or reflection; 4) AttH [4], a combination of RotH and RefH by attention mechanism; 5) UltraE [27], a method of pseudo-orthogonal transformation.

## 6.3 Results in Low Dimensions and Discussion

To assess the effectiveness of our approach, we initially evaluate it in a low-dimensional setting with $d = 32$, comparable to prior KGE methods. Table 2 presents a comparative analysis of UniGE against various baselines, including recent Euclidean and hyperbolic KGE methods. Notably, UniGE excels in providing superior representations for knowledge graph data, accommodating multiple geometric structures even in lower dimensions.

Our findings reveal that UniGE outperforms both previous Euclidean and hyperbolic KGE methods, showcasing substantial improvements. Specifically, UniGE exhibits a noteworthy performance gain of 0.6%, 1.5%, and 1.7% in terms of MRR on WN18RR, FB15k-237, and YAGO3-10, respectively. Moreover, on FB15k-237, UniGE stands out with a remarkable 9.6% performance boost compared to Euclidean KGE methods. This impressive result can be attributed to the presence of multiple hierarchical relationships in all datasets, a challenging factor in knowledge graph representation. It's worth noting that UniGE's unified geometric approach effectively captures hierarchical and chain structures, compensating for the limitations of previous Euclidean and hyperbolic KGE methods.

Additionally, we observe that UniGE surpasses RotH by 4.0% and 9.6% in MRR on WN18RR and FB15k-237, respectively. Analyzing the supplementary material, we note that WN18RR exhibits a lower $\xi_G$ value, suggesting a more tree-like structure compared to YAGO3-10. Conversely, FB15k-237 is characterized by a more prominent chain structure within the knowledge graph data. These observations corroborate the earlier results, emphasizing the significant contributions of UniGE to KGE methods operating within a single embedding space.

For methods operating in hyperbolic space, our findings demonstrate that the extent of chain structure within the knowledge graph correlates with the need for information from the Euclidean space as an auxiliary source. Furthermore, our experimental results underscore that FB15k-237, with its greater dependence on Euclidean space information, benefits significantly from our approach. This further attests to the effectiveness of UniGE in addressing the limitations of single embedding space models and improving knowledge graph representations, especially in low-dimensional settings.

## 6.4 Results in High Dimensions and Discussion

In Table 3, we present the results of link prediction in high dimensions (specifically, $d \in 200, 400, 500$). In this setting, UniGE competes against several other models and remarkably achieves new state-of-the-art results on WN18RR, YAGO3-10, and FB15k-237.

As expected, it's worth noting that UniGE consistently outperforms all the compared approaches, with the sole exception of UltraE, which achieves competitive results. Nonetheless, the performance gain observed in high-dimensional cases is less pronounced than in lower dimensions. This reinforces the idea that KG embeddings become less sensitive to embedding space as dimensionality increases.

However, when working in high dimensions, UniGE and other KGE methods exhibit similar performance across all datasets. This behavior is consistent with the observation that, in sufficiently high-dimensional spaces, both Euclidean and hyperbolic embeddings can effectively capture the complexity of hierarchies in knowledge graphs. It suggests that the choice of embedding space becomes less critical when the dimensionality is sufficiently large.

Moreover, it's plausible that the additional performance gain can be attributed to the flexibility of UniGE's inference mechanisms, which contribute to its effectiveness in capturing relational patterns in knowledge graphs.

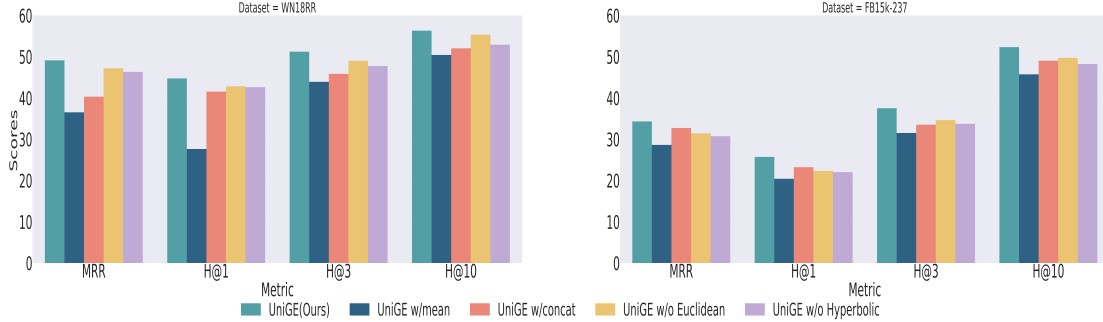

**Figure 3: Ablation Study results on WN18RR (left) and FB15k-237 (right) datasets.**

## 6.5 Performance per Relation

To assess the versatility of UniGE in modeling various relation types, we present a summary of Hits@10 for each relation within the WN18RR dataset. This analysis helps validate the exceptional representational capabilities of UniGE. We also investigate the influence of relation types on the performance of our proposed method, as well as the strong baseline GIE.

To characterize each relation, we employ two crucial metrics: the global graph curvature $\xi_G$ and the Krackhardt hierarchy score (Khs). These metrics provide insights into the presence of a rich hierarchical structure within the dataset. The curvature estimate captures global hierarchical patterns, reflecting the extent to which the graph exhibits tree-like structures when zooming out. On the other hand, the Khs characterizes more localized behaviors, specifically quantifying the number of small loops present within the graph. A lower value of $\xi_G$ indicates a higher degree of tree-like structure within the KG. In fully observed symmetric relations, Khs equals 0, and for anti-symmetric relations, Khs equals 1. Subsequently, we calculate the average Hits@10 over 10 iterations for models with low dimensionality.

The results in Table 4 clearly demonstrate that our proposed UniGE model excels in handling diverse relation types, indicating its effectiveness in addressing intricate structures within knowledge graphs (KGs). Furthermore, UniGE consistently outperforms the baseline approach GIE in modeling these relations, highlighting its robustness and superior performance across various relation types.

## 6.6 Ablation Study

In this subsection, we delve into the effectiveness of the unified embedding space within UniGE. As depicted in Figure 3, we conduct a series of experiments to examine the contributions of various components in UniGE. Specifically, we explore alternative fusion methods, replacing the optimal transport procedure with mean and concat operations, denoted as UniGE w/ mean and UniGE w/ concat, respectively. Additionally, we conduct experiments on KGE using only Euclidean or hyperbolic spaces, denoted as UniGE w/o Euclidean and UniGE w/o Hyperbolic.

The results of these ablation experiments are notably inferior to the original UniGE, underscoring the advantages of our approach. UniGE's fusion method stands out as the most effective in modeling logical patterns within the knowledge graph. In particular, the performance of concat and mean operations lags behind, reinforcing

the superiority of our proposed approach in addressing the intricate structures and relationships within KGs.

**Table 4: Comparison of hits@10 for WN18RR. GIE represents GIE in [3]**

| relation name | $\xi_G$ | Khs | GIE | UniGE |
|---|---|---|---|---|
| also_see | -2.09 | 0.24 | 0.759 | **.768** |
| hypernym | -2.46 | 0.99 | 0.262 | **.274** |
| has_part | -1.43 | 1 | 0.334 | **.341** |
| member_meronym | -2.90 | 1 | **0.360** | 0.357 |
| synset_domain_topic_of | -0.69 | 0.99 | 0.435 | **.446** |
| instance_hypernum | -0.82 | 1 | 0.501 | **.511** |
| member_of_domain_region | -0.78 | 1 | 0.404 | **.437** |
| member_of_domain_usage | -0.74 | 1 | 0.438 | **0.446** |
| derivationally_related_form | -3.84 | 0.4 | **0.968** | 0.964 |
| similar_to | -1.00 | 0 | **1** | **1** |
| verb_group | -0.5 | 0 | **0.984** | 0.981 |

## 7 CONCLUSION

In this study, we have introduced UniGE, a novel and highly effective model for KGE. To tackle the challenge of handling hybrid geometric spaces, UniGE transfers Euclidean and hyperbolic spaces into a unified space using optimal transport and fuses the two space information via the Wasserstein barycenter. Theoretical analysis supports the effectiveness of our method, with provable error bounds for the unified embeddings. Empirically, UniGE outperforms many previous Euclidean and non-Euclidean models on three standard KG datasets.

The remarkable performance of UniGE encourages further research into unified geometric approaches, not only for KGE but also for other tasks that stand to benefit from this innovative perspective, such as graph neural network-based methods. Moreover, UniGE can be extended to fuse embeddings from different types of KGs, harnessing their unique characteristics to achieve superior performance.

## 8 ETHICS STATEMENT

In this study, we utilized three publicly available KG datasets. All of these datasets are widely used in the KG research community and are free from ethical or copyright concerns.

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

# A APPENDIX

## A.1 Datesets Details

In this study, we employ three established benchmarks for evaluation: WN18RR [2], a subset of WordNet comprising 11 lexical relations; FB15k-237 [2], a Freebase subset encompassing general world knowledge; and YAGO3-10 [16], a YAGO3 subset that details relationships between individuals. We adopt the global graph curvature metric [8], in accordance with prior research [4], to assess the geometric attributes of these datasets. The datasets' characteristics are consolidated in Table 5. Our analysis reveals that all datasets exhibit a global hierarchical nature, as evidenced by their negative curvature values. However, none of the datasets strictly adhere to a tree structure. Notably, WN18RR demonstrates a higher degree of hierarchy than FB15k-237 and YAGO3-10, as indicated by its relatively lower global graph curvature.

**Table 5: The statistics of KGs,where $\xi_G$ measures the treelikeness (the lower the $\xi_G$ is, the more tree-like the KG is).**

| Dataset | $\xi_G$ | entities | relations | triples |
|---|---|---|---|---|
| WN18RR | -2.54 | 41k | 11 | 93k |
| FB15k-237 | -0.65 | 15k | 237 | 310k |
| YAGO3-10 | -0.54 | 123k | 37 | 1M |

