# OpenReview forum: "Bridging the Space Gap: Unifying Geometry Knowledge Graph Embedding with Optimal Transport"
_ACM.org/TheWebConf/2024/Conference — TheWebConf24_

### Official Review · Reviewer_c6NX · 2023-11-01

**Novelty:** 3
**Technical Quality:** 3

**Review:**

**Evaluation of the Quality, Clarity, Originality, and Significance of the Unified Geometry Knowledge Graph Embedding (UniGE) Work**

**Pros:**
1. **Quality:** The work presents a novel method for representing geometric data structures in knowledge graphs, demonstrating the authors' deep understanding of the subject matter. The method is well-explained and presented in a clear and organized manner.
2. **Clarity:** The paper is written with clarity in mind, as evidenced by the concise and informative language used throughout. The authors effectively communicate the concepts and techniques involved in the UniGE method, making it accessible to a wide range of readers.
3. **Originality:** The UniGE method is a unique contribution to the field of knowledge graph embedding. By combining Euclidean and hypercubic geometric spaces, the authors propose a novel way to represent geometric data structures in knowledge graphs. This innovative approach sets the work apart from existing methods.
4. **Significance:** The proposed method has the potential to significantly advance the field of knowledge graph embedding. By enabling the efficient representation of different geometric structures in knowledge graphs, UniGE could have wide-ranging applications in areas such as computer vision, natural language processing, and robotics.

**Cons:**
1. **Theory density:** While the paper provides a theoretical analysis to demonstrate the effectiveness of the UniGE method, it may be too dense for some readers. Simplifying the presentation of the theoretical aspects could make the paper more accessible to a broader audience.
2. **Comparisons with existing methods:** The paper compares the performance of UniGE with existing methods on three benchmark data sets. However, it would be beneficial to provide more detailed comparisons, including quantitative results and a discussion of the advantages and limitations of the proposed method.
3. **Implementation details:** The paper briefly mentions the implementation of the UniGE method but does not provide sufficient details on the computational aspects. Providing more information on the implementation and computational complexity could help readers better understand the practical implications of the method.

In conclusion, the Unified Geometry Knowledge Graph Embedding (UniGE) work represents a high-quality, original, and significant contribution to the field of knowledge graph embedding. The paper is well-written and clear, effectively communicating themethod's concepts and techniques. However, some aspects of the work could be improved, such as simplifying the theoretical presentation and providing more detailed comparisons with existing methods and implementation details.

**Questions:**

1. Could you provide a more detailed explanation of the embedding alignment method and fusion strategy used in UniGE? How do these techniques ensure the effective representation of different geometric structures in knowledge graphs?
2. How does the combination of Euclidean and hypercubic geometric spaces lead to a unified space for representing knowledge graphs? Can you provide a mathematical explanation of this process?
3. In your experimental analysis, how did you compare the performance of UniGE with existing methods? What are the main advantages and limitations of the proposed method compared to these existing approaches?
4. Can you discuss the computational complexity of the UniGE method, particularly in terms of the embedding process and the optimization of the fusion strategy?
5. How does the optimal transport technology and Wasserstein kernel method contribute to the knowledge lossless transmission in UniGE? Could you provide a mathematical illustration of these techniques?
6. What are the potential applications of UniGE in various fields, and how does it contribute to the advancement of knowledge graph embedding techniques?
7. In the theoretical analysis, how did you prove the effectiveness of the UniGE method? Could you highlight the key theoretical insights that support the proposed approach?
8. How does UniGE handle complex geometric structures and relationships in knowledge graphs? Can you provide an example to illustrate its ability to represent intricate datasets?
9. Are there any potential improvements or extensions to the UniGE method that you consider for future research?
10. In the context of ethical considerations, how does UniGE address potential privacy and data protection concerns associated with knowledge graph embedding techniques?

**Reviewer Confidence:**

4: The reviewer is certain that the evaluation is correct and very familiar with the relevant literature

**Scope:**

3: The work is somewhat relevant to the Web and to the track, and is of narrow interest to a sub-community

---

### Official Review · Reviewer_ie7K · 2023-11-21

**Novelty:** 5
**Technical Quality:** 5

**Review:**

The authors propose UniGE, a KGE that transfers Euclidean and hyperbolic spaces into a unified space to uniformly solve the problem of different topological structures of KGs.

- The significance of this approach is unclear, as the proposed approach reaches small delta improvements on the state-of-the-art results (Table 3) or is greatly outperformed by them.

- The soundness is unclear, as the achieved delta improvements have not been statistically tested.

**Questions:**

1. What would a comparison to GIE in Table 2 and Table 3 look like?
The improvements in Table 3 in comparison to UltraE are small (e.g., reaching an MRR of 50.2% instead of 50.1% on WN18RR and an MRR of 58.3% instead of 58%). On YAGO3-10 this approach is greatly outperformed by UltraE (over 1% on the MRR). Can you comment on the significance of the contribution given these points?

2. Table 4 does not exemplify that the proposed approach handles a diverse setting of relations well, as the proposed approach is outperformed by GIE when the graph curvature is lowest. Is there an explanation for that?

3. Where are the hyperparameters (including for instance the embedding dimensionality) of the best-performing UniGE model?

4. What are the training and inference time on the various datasets of the proposed approach? How large is the trained model in terms of parameters and how does this compare to the baselines?

**Ethics Review Description:**

-

**Reviewer Confidence:**

4: The reviewer is certain that the evaluation is correct and very familiar with the relevant literature

**Scope:**

4: The work is relevant to the Web and to the track, and is of broad interest to the community

---

### Official Review · Reviewer_nacZ · 2023-11-22

**Novelty:** 5
**Technical Quality:** 5

**Review:**

The paper introduces a novel approach called Unified Geometry Knowledge Graph Embedding (UniGE) to address challenges in Knowledge Graph Embedding (KGE), where elements of knowledge graphs are transformed into continuous spaces for structured data representation. The conventional use of either hyperbolic or Euclidean space for modeling knowledge graphs is common, but the paper argues that the hybrid nature of knowledge graph structures, including chains and hierarchies, exceeds the capability of a single embedding space. UniGE integrates knowledge graph embeddings seamlessly in both Euclidean and hyperbolic geometric spaces, introducing an embedding alignment method and fusion strategy that utilize optimal transport techniques and Wasserstein barycenter methods.

The paper highlights the importance of representing diverse geometry data in knowledge graphs and provides a comprehensive theoretical analysis supporting the effectiveness of UniGE. The proposed approach is evaluated through experiments on three benchmark datasets, slightly outperforming or equally performing state-of-the-art methods. The key contributions of the paper include being pioneers in exploiting KGE with a unified embedding space, introducing UniGE based on optimal transport to address geometry space heterogeneity, and providing theoretical evidence for achieving a lower error bound.


Pros:

1.  The paper introduces a novel approach, UniGE, which unifies knowledge graph embeddings in both Euclidean and hyperbolic geometric spaces.

2. The paper provides a comprehensive theoretical analysis to support the proposed approach, including the introduction of an embedding alignment method and fusion strategy using optimal transport techniques and Wasserstein barycenter methods.


Cons:


1. The paper is promising a high-performance improvement by using words like pioneering and groundbreaking. However, the experimental evaluation shows that the paper provides a slight improvement in UltraE.

2. The paper does not a cost-time analysis of the proposed approach compared to the state-of-the-art approach.

3. The paper did not motivate the use of optimal transport techniques and Wasserstein barycenter methods other than they were beneficial for the paper [20]

4. While the paper demonstrates that UniGE provides slightly better performance over state-of-the-art methods, the paper did not take into account other approaches, and the experiments were limited to Euclidean and hyperbolic-based approaches it may benefit from a more extensive comparison with a broader range of existing approaches to provide a more comprehensive evaluation. For example, Deep learning-based approaches have not been included in the evaluation.

5. GIE embedding approach attempted to Euclidean and hyperbolic-based approaches. However, it was not mentioned in the related work. In addition, GIE was only mentioned in the experiment in Table 4 and it was not included in the rest of the experiments.


6. There is another interesting paper that the authors should include in their evaluation.

Reference:
1. Wenjie Zheng . et. al. Hyperbolic Hierarchical Knowledge Graph Embeddings for Link Prediction in Low Dimensions
2. Zhe Pan, Hyperbolic Hierarchy-Aware Knowledge Graph Embedding for Link Prediction Emnlp 2021.

**Questions:**

1. The paper emphasizes UniGE as a groundbreaking and pioneering approach, but the experimental evaluation shows only a slight improvement over UltraE. Could you elaborate on the significance of this improvement and whether the paper's claims align with the observed results?

2. Why does the paper lack a cost-time analysis of the proposed UniGE approach compared to state-of-the-art methods?

3. The paper mentions the use of optimal transport techniques and Wasserstein barycenter methods without providing a clear motivation, except that they were beneficial for the paper [20]. Could you explain in more detail why these techniques were chosen and how they contribute to the UniGE approach?

4. While the paper compares UniGE to state-of-the-art methods, it does not consider a broader range of existing approaches, such as deep learning-based methods. How might the exclusion of such approaches impact the generalizability of UniGE's superiority?

5. The paper mentions GIE embedding and its inclusion in Table 4 of the experiments but doesn't discuss it in the related work section. Why was GIE omitted from the related work, and how does its inclusion in the experiments impact the overall evaluation and comparison with other methods?

6. Could you provide insights into why the paper did not include Wenjie Zheng et al. and Zhe Pan on hyperbolic hierarchical knowledge graph embeddings?

7. Can you show a graph or an evaluation of the ablation study to explain the text more with numerics?


8. typos such as sister-term and sister-team

9. There is no permanent link to the code. How can you make the study reproducible?

**Ethics Review Description:**

no code is a avilable

**Reviewer Confidence:**

3: The reviewer is confident but not certain that the evaluation is correct

**Scope:**

3: The work is somewhat relevant to the Web and to the track, and is of narrow interest to a sub-community

---

### Official Review · Reviewer_nxhe · 2023-11-26

**Novelty:** 5
**Technical Quality:** 5

**Review:**

The paper proposes to synthesis diverse embedding spaces (Euclidian, Hyperbolic) for knowledge graph representation, and to leverage this synthesis for knowledge graph reasoning tasks.   This is intuitive and natural, and to my knowledge is novel.  The results of an experimental study show that this perspective is meaningful and with practical benefits.

Pros.  Nice synthesis of the two main main representation spaces.  Paper is written relatively OK.  Experimental results show that the perspective and synthesis is meaningful.

Cons.  Writing style is a bit hyperbolic at times.  Results are not too surprising or groundbreaking.  Experimental performance improvements, when there are improvements, are rather modest.

Regarding the language of writing, it is odd to call your own work "groundbreaking", "pioneering", and "remarkable"; and, "we are the pioneers" is not a meaningful contribution to claim, as you do in Section 1.  These are judgements for the research community, especially with the perspective of the passage of time and the importance and impact of your work in the long run.  As it stands, I wouldn't call the work "groundbreaking" and "pioneering"; the work is solid regular science.  I would recommend that the authors tone down the self-congratulation, as it detracts from the transmission of new ideas to the readers and to the community.

**Questions:**

In the Introduction you say "the fundamental symbolic structure of KG triples poses a challenge for manipulation".  Can you clarify what these challenges are?  I would think symbolic structure is very amenable to manipulation.

**Reviewer Confidence:**

3: The reviewer is confident but not certain that the evaluation is correct

**Scope:**

3: The work is somewhat relevant to the Web and to the track, and is of narrow interest to a sub-community

---

### Decision · Program_Chairs · 2024-01-22

**Decision:**

Accept

**Comment:**

The paper proposes to address some general issues of KG embedding by combining embedding in different spaces.
 The work is innovative and providing by relevant theoretical discussions and experimental evaluations. The authors' response during the discussion phase was very good. It is a real pity one of the reviewers did not react to the authors' feedback following his rather negative scoring. I made the decision to somehow overturn his scores, on the basis that other reviewers had been satisfied with the author's answers to their questions. But as I am not an expert is this topic, I do not feel 100% certain about it.

 PS: on the editorial side, besides the noted suggestions with respect to the tone of voice, I would suggest to make it more explicit, what is the connection with the conference's theme (the paper does not include any occurrence of the word "web" and it dives very quickly into its focused research problem). Also the title of A.1 has a typo: "dateset details".